# Rational Design of Potent α-Conotoxin PeIA Analogues with Non-Natural Amino Acids for the Inhibition of Human α9α10 Nicotinic Acetylcholine Receptors

**DOI:** 10.3390/md22030110

**Published:** 2024-02-27

**Authors:** Tianmiao Li, Han-Shen Tae, Jiazhen Liang, Zixuan Zhang, Xiao Li, Tao Jiang, David J. Adams, Rilei Yu

**Affiliations:** 1Key Laboratory of Marine Drugs, Chinese Ministry of Education, School of Medicine and Pharmacy, Ocean University of China, 5 Yushan Road, Qingdao 266003, China; litianmiao99@163.com (T.L.); 17863973220@163.com (J.L.); zzx7212@stu.ouc.edu.cn (Z.Z.); 17854203983@163.com (X.L.); jiangtao@ouc.edu.cn (T.J.); 2Laboratory for Marine Drugs and Bioproducts, Qingdao National Laboratory for Marine Science and Technology, Qingdao 266003, China; 3Molecular Horizons, Faculty of Science, Medicine and Health, University of Wollongong, Wollongong, NSW 2522, Australia; hstae@uow.edu.au; 4Innovation Center for Marine Drug Screening & Evaluation, Qingdao National Laboratory for Marine Science and Technology, Qingdao 266003, China

**Keywords:** α-conotoxin, PeIA, nicotinic acetylcholine receptor, non-natural amino acid, molecular dynamics simulations

## Abstract

α-Conotoxins (α-CTxs) are structurally related peptides that antagonize nicotinic acetylcholine receptors (nAChRs), which may serve as new alternatives to opioid-based treatment for pain-related conditions. The non-natural amino acid analogues of α-CTxs have been demonstrated with improved potency compared to the native peptide. In this study, we chemically synthesized Dab/Dap-substituted analogues of α-CTx PeIA and evaluated their activity at heterologously expressed human α9α10 nAChRs. PeIA[S4Dap, S9Dap] had the most potent half-maximal inhibitory concentration (IC_50_) of 0.93 nM. Molecular dynamic simulations suggested that the side chain amino group of Dap4 formed additional hydrogen bonds with S168 and D169 of the receptor and Dap9 formed an extra hydrogen bond interaction with Q34, which is distinctive to PeIA. Overall, our findings provide new insights into further development of more potent analogues of α-CTxs, and PeIA[S4Dap, S9Dap] has potential as a drug candidate for the treatment of chronic neuropathic pain.

## 1. Introduction

Chronic pain is a significant problem that threatens human health worldwide, but opioid-based medications can lead to drug tolerance and addiction. Thus, non-opioid therapeutics are urgently needed [1,2]. Previous studies have shown that several subtypes of nAChRs are associated with pain and may serve as new treatment targets [3].

nAChRs belong to the Cys-loop superfamily and are prototypical characterized members of pentameric ligand-gated ion channels (LGICs) assembled from homologous subunits (α1–α10, β1–β4, δ, ε, and γ,) [4]. The muscle-type nAChRs consist of α1, β1, δ, and ε/γ subunits, whereas other subunits form distinct homomeric or heteromeric nAChRs, such as α7, α3β2, α3β4, and α9α10 [5]. The diverse subunit compositions of nAChRs give rise to many subtypes with differences in ion channel function and pharmacology. The two adjacent cysteines in loop C of the α subunit are essential for ACh to activate nAChRs, and the binding sites of nAChRs are located at the interface of two adjacent α and α/β subunits [5].

There are several nAChR subtypes associated with pain, including α3-containing nAChRs such as α3β2 and α3β4 that are expressed in primary afferent nerve terminals or the spinal cord, which may mediate anti-allodynic actions [6]. Additionally, α6β4-containing nAChRs may be involved in the processing of pain-associated information and activation of these nAChRs could produce analgesic effects via purinergic (PX2) receptor inhibition [7]. The α7 subtype is widely distributed in the nervous system and expressed by many non-neuronal cells (microglial, lymphocytes, and macrophage cells), and targeting α7 nAChRs has proven to be effective in attenuating neuropathic and inflammatory pain in animal models [8]. The α9α10 nAChRs (Figure 1A,B) have been studied extensively and the analgesic effects of several α9α10 nAChR antagonists have been demonstrated in animal models for chronic constriction injury [9,10], oxaliplatin-induced neuropathy [11], and partial nerve ligation [10]. The vital functions of α9α10 nAChRs in modulating the pathophysiology changes associated with traumatic and nerve injury [12] provide an initial basis for the development of non-opioid analgesic drugs that target these receptors.

Conotoxins (CTxs) derived from the venom of *Conus* marine snails are disulfide-rich peptides for rapid prey immobilization [13]. Among them, α-CTxs are the most characterized and studied family, which are structurally related peptides that target nAChRs [14]. The α-CTxs consist of 12~20 residues with four cysteines that form two disulfide bridges to provide conformation stability as well as interactions with receptors [15,16]. The high affinity and selectivity of α-CTxs towards particular nAChR subtypes, coupled with low toxicity, give them an advantage as potential therapeutic candidates [17].

α-CTx PeIA (Figure 1C,D) from the venom of *C. pergrandis* is relatively potent at inhibiting human (h) α9α10 nAChRs [18,19]. Interestingly, PeIA is structurally similar to α-CTxs Vc1.1 and Mr1.1. Our previous research demonstrated that the replacement of residue Ser4 of Vc1.1 with the non-natural amino acid Dab (2,4-diaminobutanoic acid) and residues Ser4/Ser9 of Mr1.1 with the non-natural amino acid Dap (2,3-diaminopropionic acid) (Appendix A) significantly increased potencies of the analogues at inhibiting hα9α10 nAChRs [9,20]. Thus, we hypothesized the Dab/Dap-incorporated analogues of PeIA may also improve the potency of the parent peptide via extra interactions with the receptor.

We chemically synthesized four non-natural amino acid analogues, PeIA[S4Dab], PeIA[S4Dap], PeIA[S9Dap], and PeIA[S4Dap, S9Dap], and evaluated their activity at hα9α10 nAChRs heterologously expressed in *Xenopus laevis* oocytes using the two-electrode voltage clamp technique. Additionally, we established binding models of the most potent analogue, PeIA[S4Dap, S9Dap], with hα9α10 nAChRs to better understand the binding mode of the key Dap residue interacting with the receptor. In summary, the IC_50_ values of PeIA[S4Dab], PeIA[S4Dap], PeIA[S9Dap], and PeIA[S4Dap, S9Dap] were enhanced by ~4-, 13-, 5-, and 24-fold, respectively, relative to PeIA. Molecular dynamics (MD) simulations suggested the amino group of the Dap4 side chain formed three hydrogen bonds with α9(+)-α9(−) interface residues D166, S168, and D169 of the α9(−) subunit, as opposed to one hydrogen bond between the PeIA S4 residue with D169. Additionally, Dap9 formed a hydrogen bond interaction with Q34 of the α9(−) subunit. Taken together, these additional interactions may account for the improved potency of the analogues at inhibiting the hα9α10 nAChR.

## 2. Results

### 2.1. Design and Synthesis of PeIA Analogues

Based on the same structural subclass and the sequence similarity of PeIA, Vc1.1, and Mr1.1 (Figure 2A), in addition to similarities between the sequence in Loop 1 and the charge distribution in Loop 2 of PeIA and Mr1.1, we attempted to apply this strategy to PeIA by the substitution of Ser4 with Dab or Dap, Ser9 with Dap, or both Ser4 and Ser9 with Dap (Figure 2B). The peptides were synthesized by solid-phase peptide synthesis (SPPS) on Rink-Amide-MBHA resin, followed by peptide cleavage from the resin (Figure 2C). Regioselective oxidation was performed at CysI-CysIII and CysII-CysIV, and the Cys was introduced in pairs with Fmoc-Cys(Acm)-OH and Fmoc-Cys(Trt)-OH. Finally, the peptide was purified by preparative high-performance liquid chromatography (HPLC) with C_18_ columns, followed by mass spectrometry (MS) and analytical HPLC to verify the molecular mass and purity of the peptides (Appendix A).

### 2.2. Structural Analysis of the Designed PeIA Analogues

Circular dichroism (CD) spectroscopy was used to characterize the secondary structures of the designed analogues and PeIA (Figure 3). The four analogues exhibited significant negative absorption at 208 nM, similar to PeIA. The results indicated that the Dab/Dap-substituted analogues had minimal impact on the secondary structure of PeIA.

### 2.3. Activity of PeIA Analogues at Heterologous Human nAChRs

The activity of PeIA analogues was evaluated on ACh-evoked currents mediated by hα9α10 nAChRs heterologously expressed in *X. laevis* oocytes using the two-electrode voltage clamp technique. PeIA[S4Dab], PeIA[S4Dap], PeIA[S9Dap], and PeIA[S4Dap, S9Dap] reversibly inhibited α9α10 ACh-evoked currents in a concentration-dependent manner, giving IC_50_ values of 5.32 nM, 1.74 nM, 4.67 nM, and 0.93 nM, respectively (Figure 4 and Appendix A). In comparison to PeIA, which has an IC_50_ value of 21.9 nM [19], the activities were enhanced ~4-, 13-, 5-, and 24-fold, respectively. The enhanced potency effect of Dab/Dap substitution on PeIA is consistent with our previous studies on Vc1.1 and Mr1.1 [9,20].

The subtype selectivity of PeIA and the most potent analogue, PeIA[S4Dap, S9Dap], was determined for different human nAChR subtypes. Both PeIA and PeIA[S4Dap, S9Dap] (10 and 100 nM) were inactive (<1% inhibition) on ACh-evoked currents mediated by hα1β1εδ, hα1β1δγ, hα3β4, hα4β2, and hα4β4 nAChR subtypes (Figure 5A,B). At hα3β2 nAChRs, PeIA was ~2-fold less potent (IC_50_ = 59.8 nM) than PeIA[S4Dap, S9Dap] (IC_50_ = 32.8 nM) (Figure 5C,D). Although PeIA was inactive at hα7 (Figure 4A), PeIA[S4Dap, S9Dap] inhibited hα7 nAChRs with an IC_50_ of 161.7 nM (Figure 5D). 

### 2.4. Molecular Dynamic (MD) Simulations of PeIA and Its Analogue in Complex with Human α9α10 nAChRs

To explain the significantly improved potencies of the most potent analogue PeIA[S4Dap, S9Dap], we established the complex structure model of the peptide and hα9α10 nAChR using MD simulations (Figure 6 and Appendix A) based on the models of Mr1.1 and Vc1.1 [9,20]. The Ser residue at position 4 in PeIA possesses a hydroxyl side chain, whereas the substituted Dap residue contains an amino side chain. The amino group formed three hydrogen bonds with D166, S168, and D169 at the α9(+)-α9(−) interface, whereas S4 of PeIA only had one hydrogen bond with D169. Moreover, Dap9 formed a hydrogen bond interaction with Q34 in the α9(-) subunit. The formation of extra hydrogen bonds as well as the electrostatic interactions between Dap and residues at the α9(+)-α9(−) are postulated to contribute to the significantly enhanced activity of PeIA[S4Dap, S9Dap] compared to the native PeIA. 

## 3. Discussion

CTxs are pharmacologically valuable peptides with selectivity for different nAChR subtypes. Several α-CTxs, including PeIA [18], RgIA [21], Vc1.1 [22], and Mr1.1 [9], are antagonists of α9α10 nAChRs, and the S4 residue is conserved in these α-CTxs (Figure 2A). In addition, their structure–activity relationship at the α9α10 subtype has also been investigated. The S4 residue of Vc1.1 is proposed to be in close proximity to the α9 subunit D166 and D169 residues. Both S4Dab and S4Dap substitutions in Vc1.1 remarkably increased the peptide potency, whereas the [S4K] mutation decreased its potency. Thus, a basic residue with an appropriate side chain length is crucial to the potency of these mutants [20]. Similarly, Dap-substituted analogues of Mr1.1, Mr1.1[S4Dap], Mr1.1[S9Dap], and Mr1.1[S4Dap, S9Dap], showed significantly enhanced potency at α9α10 nAChRs compared to Mr1.1 [9].

At the heterologous human nAChR subtypes tested, PeIA is most selective for hα9α10 followed by the hα3β2 subtype, and PeIA[S4Dap, S9Dap] retains the α9α10-selectivity of PeIA. Remarkably, compared to the absence of inhibition by PeIA, the analogue has enhanced potency at hα7 nAChRs. Despite this, it has been proposed that activation of the α7 subtype, as opposed to inhibition, mediates the attenuation of neuropathic pain in animal models [23,24,25]. For hα9α10 nAChRs, the ~24-fold increase in potency for PeIA[S4Dap, S9Dap] compared to the native PeIA resulted in one of the most potent hα9α10 antagonists. To date, only two analogues of RgIA, RgIA-5524 and RgIA-5474, have been reported to antagonize the hα9α10 subtype with IC_50_ values below 1 nM (0.9 and 0.05 nM, respectively) [26,27]. Previously, substantial efforts have been made to optimize the structure of PeIA. Liang et al. [19] engineered dimeric PeIA with an 11-fold enhanced activity and an IC_50_ of 1.9 nM, presumably via simultaneous interactions with neighboring binding sites on the receptor. Taken together, the potency of PeIA[S4Dap, S9Dap] is approximately two-fold higher than dimeric PeIA, and the synthesis procedure for the former is more facile. Thus, PeIA[S4Dap, S9Dap] possesses an advantage as a potential drug candidate in the treatment of pain-related conditions.

## 4. Materials and Methods

### 4.1. Peptide Synthesis

Peptides were synthesized by solid-phase peptide synthesis (SPPS). Rink amide MBHA resin (0.1 mmol; SV = 0.6 mmol/g) was used as a carrier for amino acid coupling for the amidated C-terminus of the peptide. There were 4 cysteine residues in the peptide, and an acid-labile trityl (Trt) protecting group was introduced in Cys2 and Cys8, while Cys3 and Cys16 were protected by acid-stable acetamidomethyl (Acm) to realize orthogonal oxidation. The specific experimental steps were described as follows: 166 mg of resin was put in a reaction vessel and swelled using a mixture solution of DMF/DCM (1:1). Before coupling, a 20% piperidine solution in DMF was used to deprotect the Fmoc (9-fluorenylmethyloxycarbonyl) protecting group. The carboxyl group of the newly introduced N-Fmoc-protected residue was coupled with the amino group of the resin using HCTU (4.0 equiv) and DIPEA (8.0 equiv) at room temperature for an hour to realize peptide synthesis. The deprotection and coupling procedures were repeated until the full sequence was synthesized, and the terminal Fmoc group was removed. Then, a cleavage cocktail (TFA/TIPS/H_2_O, 90/5/5, *v*/*v*/*v*) was added to cleave the peptide for 3 h. After sufficient reaction, the solution was filtrated and then concentrated, precipitated with cold ether, and centrifuged to obtain the synthesized peptide. The molecular weight of the crude peptide was detected by Electrospray Ionization Mass Spectrometry (ESI-MS) (Appendix A).

### 4.2. Disulfide Bond Formation between Cys2 and Cys8

The Trt group was removed along with peptide cleavage. The crude peptide was dissolved in a mixture of H_2_O/MeCN, and 1 equiv of 2,2′-dithiodipyridine (DTDP) in MeOH was added at a slow rate. The solvent was then stirred at room temperature for 1 h.

### 4.3. Disulfide Bond Formation between Cys3 and Cys16

I_2_-mediated oxidation was used to form the second disulfide bond. The purified peptide was dissolved in H_2_O/MeCN (0.5 mg/mL), and superfluous I_2_ in MeCN was added until the solution turned yellowish brown. After stirring at room temperature for 3 h, the reaction was quenched by ascorbic acid.

### 4.4. Peptide Purification

Semipreparative reverse-phase high-performance liquid chromatography (RP-HPLC) with a C_18_ column was carried out in the purification assay. Deionized H_2_O and chromatographic grade MeCN were the components of the moving phase. The H_2_O/MeCN gradient at a flow rate of 6 mL/min containing 0.05% trifluoroacetic acid (TFA) is shown in Table 1. The analytical RP-HPLC used to determine the purity of the peptide and the H_2_O/MeCN gradient are shown in Table 1.

### 4.5. Circular Dichroism

Circular dichroism (CD) spectra were performed on Jasco J-810 spectropolarimeter over a wavelength range of 250–190 nm using a 1.0 mm path length cell, a bandwidth of 1.0 nm, a response time of 2 s, and averaging over three scans. Spectra were recorded at room temperature under a nitrogen atmosphere. Peptides were dissolved in acetonitrile and H_2_O (1:1) and depicted as molar ellipticity ([θ]): [θ] = 1000·mdeg/(*l* × *c*), where mdeg is the raw CD data, *c* is the peptide molar concentration (mM), and *l* is the cell path length (mm). All peptides were measured at a concentration of 0.2 mg/mL.

### 4.6. Xenopus laevis Oocyte Preparation and Microinjection

The protocols were approved by the Animal Ethics Committees (project no. AE2003) of the University of Wollongong. The human α1, β1, δ, ε, and γ nAChR clones were obtained from Integrated DNA Technologies (Coralville, IA, USA) and human α3, α9, α10, β2, and β4 nAChR clones were obtained from OriGene (Rockville, MD, USA) and sub-cloned into the pT7TS vector. The hα7 (J. Lindstrom, University of Pennsylvania, USA) and hα4 clones are in plasmid pMXT and plasmid pSP64, respectively. Constructs of human nAChR subunits were linearized for in vitro cRNA transcription using the SP6/T7 mMessage mMachine^®^ transcription kit (AMBION, Forster City, CA, USA).

A maximum of 4 female *X. laevis* were kept in a 15 L aquarium (20–26 °C, light and dark cycle). Stage V-VI oocytes (Dumont’s classification; 1200–1300 μm diameter) were obtained from five-year-old frogs anesthetized with 1.7 mg/mL ethyl 3-aminobenzoate methanesulfonate (pH 7.4 with NaHCO_3_) and defolliculated with 1.5 mg/mL collagenase Type II (GIBCO, Grand Island, NY, USA) at room temperature for 1 h in OR-2 solution containing (in mM): 82.5 NaCl, 2 KCl, 1 MgCl_2_, and 5 HEPES at pH 7.4.

Oocytes were injected with 5 ng of cRNAs for human α1β1εδ, α1β1δγ, α3β2, α3β4, α4β2, and α4β4 nAChRs, 10 ng of cRNAs for hα7 nAChRs, and 35 ng of cRNAs for hα9α10 nAChRs (concentrations were confirmed spectrophotometrically and by gel electrophoresis). The muscle subunit cRNA ratio was 2:1:1:1 (α1–β1–ε/γ–δ) and the heteromeric α and αβ subunit cRNA ratio was 1:1, and was injected using glass pipettes (3–000-203 GX, Drummond Scientific Co., Broomall, PA, USA). The injected oocytes were incubated in sterile ND96 solution at 18 °C composed of (in mM): 96 NaCl, 2 KCl, 1 CaCl_2_, 1 MgCl_2_, and 5 HEPES at pH 7.4, supplemented with 5% fetal bovine serum (Bovogen, East Keilor, VIC, Australia), 0.1 mg/L gentamicin (GIBCO), and 100 U/mL penicillin–streptomycin (GIBCO).

### 4.7. Oocyte Two-Electrode Voltage Clamp Recording and Data Analysis

After 2–5 days of cRNA microinjection, two-electrode voltage clamp recording was carried out on *X. laevis* oocytes expressing nAChRs at room temperature using a GeneClamp 500B amplifier and pClamp 9.2 software interface (Molecular Devices, San Jose, CA, USA) at a holding potential −80 mV. Voltage-recording and current-injecting electrodes were pulled from GC150T-7.5 borosilicate glass (Harvard Apparatus, Holliston, MA, USA) and filled with 3 M KCl, giving resistances of 0.3–1 MΩ. All except hα9α10-expressing oocytes were superfused with ND96 solution. Before recording, the oocytes expressing hα9α10 nAChRs were incubated in 100 μM BAPTA-AM (Sigma-Aldrich, St. Louis, MO, USA) for ∼3 h and superfused with ND115 solution containing (in mM): 115 NaCl, 2.5 KCl, 1.8 CaCl_2_, and 10 HEPES at pH 7.4. Due to the Ca^2+^ permeability of hα9α10 nAChRs, *X. laevis* oocytes were incubated in the presence of BAPTA-AM to prevent the activation of endogenous Ca^2+^-activated chloride channels. 

Initially, oocytes were briefly superfused with ND96/ND115 solution, and then three applications of acetylcholine (ACh) at a half-maximal effective concentration (EC_50_) (3 µM for hα4β2; 5 µM for hα1β1εδ and hα1β1δγ; 6 µM for hα3β2, hα4β4, and hα9α10; 100 µM for hα7; and 300 µM for hα3β4) were performed, followed by a 3 min washout between ACh applications. 

Oocytes were incubated with α-CTx for 5 min with the perfusion system turned off, followed by co-application of α-CTx and ACh with flowing bath solution. α-CTx was prepared in ND96/ND115 + 0.1% bovine serum albumin (Sigma–Aldrich). Clampfit version 10.7.0.3 software (Molecular Devices, San Jose, CA, USA) was used to measure peak current amplitudes before (ACh alone) and in the presence of α-CTx (ACh + α-CTx), where the ratio of the ACh + α-CTx-evoked current amplitude to ACh alone-evoked current amplitude was used to assess the activity of the peptides at nAChRs. Electrophysiological data were combined (*n* = 5–15) and represent the means ± standard deviations (SD). Data analysis was performed by GraphPad Prism 9 (GraphPad Software, version 9.1.0, La Jolla, CA, USA). α-CTx IC_50_ values were determined from concentration–response relationships fitted to a non-linear regression function and reported with 95% confidence intervals.

### 4.8. Construction of Non-Natural Amino Acid Parameters

The three-dimensional spatial structure of α-CTx PeIA and its non-natural amino acid-substituted analogues were constructed using Mr1.1 as a template. Based on the structural similarity between the two α-CTxs, the molecular and non-natural amino acids were constructed using the protein builder and the molecule builder module of MOE followed by molecular optimization. Due to only the coordinate space of 20 natural amino acids being identified, the parameters of Dap needed preparation. Here, the Antechamber module of AMBER22 was used to conduct this experiment as described previously [19]. 

### 4.9. Molecular Dynamics Simulation

The structure of α-CTx/α9α10nAChR was obtained by modification of the Ser4/Ser9 to Dap from our previous MD refined model between Vc1.1/α9α10nAChR [20]. Parameters for the non-natural amino acids were prepared in the Antechamber module of AMBER22. R.E.D Tools were used to produce the atom partial charges for pyroglutamic acid. The PeIA/α9α10nAChR complexes were solvated at a size distance of a 10 Å truncated regular octahedron TIP3P water box in AMBER22. Sodium ion was added to neutralize the whole system, and then the energy of the system was optimized. Firstly, a harmonic force with an elastic constant of 100 kcal mol^−1^·Å^−2^ was used to constrain the solute, and 3000 steps of the steepest descent method and the 3000-step conjugate gradient method were used to optimize the system. Secondly, the second round of optimization was carried out with all position restrictions withdrawn. The temperature of the whole system was gradually heated from 50 K to 300 K over 100 ps in the NVT complex, and over 100 ps and a 5 kcal mol^−1^·Å^−2^ harmonic force were used to restrain the position of the solute. In the isothermal-isobaric (NPT) ensemble, MD simulations were performed with the position restrictions gradually removed. In this work, MD simulations were performed over 50 ns with the pressure and the temperature maintained at 1 atmosphere and 300 K, respectively. Visual Molecular Dynamics (VMD) was used to analyze the trajectories and the root mean square deviation (RMSD) of the backbone was calculated.

## 5. Conclusions

The introduction of non-natural amino acids to peptides or proteins can provide unique three-dimensional spatial structures with unique functionalities. In recent years, many conotoxins containing non-natural amino acids have been engineered to improve the physicochemical properties of peptides, including activity, selectivity, and serum stability. The replacement of Asn11 with non-natural amino acids with a negatively charged side chain and α-aminopimelic acid (Api) produced the PeIA-5466 analogue with ~300-fold greater selectivity for α3β2 than α6/α3β2β3 subtypes [28]. In addition, the RgIA4 analogue was designed by substituting Arg9 and Tyr10 residues with Cit and 3-I-Try, respectively, which exhibited high potency at both human and rodent α9α10 nAChRs while retaining selectivity for the α9α10 subtype [29].

In this study, the non-natural amino acids Dab and Dap were used to substitute the Ser residue to optimize the activity of PeIA at inhibiting the hα9α10 nAChR. The IC_50_ values of the PeIA[S4Dab], PeIA[S4Dap], PeIA[S9Dap], and PeIA[S4Dap, S9Dap] were improved by ~4-, 13-, 5-, and 24-fold, respectively, compared to PeIA. To date, PeIA[S4Dap, S9Dap] is one of the most potent peptide antagonists of hα9α10 nAChRs. Additionally, we performed an MD simulation to reveal additional interactions that may explain the enhanced potencies of the designed analogues. Overall, we provided a simple and effective way to improve the potency of α-CTx PeIA, and the PeIA[S4Dap, S9Dap] analogue has therapeutic potential to be explored in the future.

## Figures and Tables

**Figure 1 marinedrugs-22-00110-f001:**
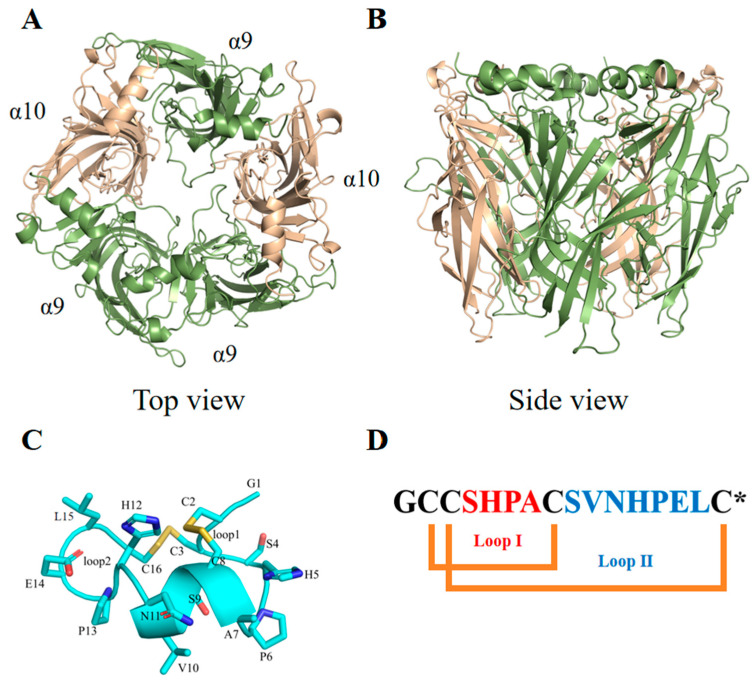
The extracellular domain of the hα9α10 nAChR and the structure of α-conotoxin PeIA. (**A**,**B**) The extracellular ligand binding domain of the hα9α10 nAChR, based on the crystal structure of *Aplysia californica* acetylcholine binding protein (AChBP) in complex with PeIA (PDB Code: 5JME), was built using Modeler and AlphaFold2 as described in our previous study [9]. (**C**,**D**) The backbone and side chains of PeIA are shown as a ribbon and sticks, respectively. α-Conotoxin PeIA is a 16-amino-acid peptide that has two disulfide bonds and an amidated C-terminus (represented by “*” in the sequence).

**Figure 2 marinedrugs-22-00110-f002:**
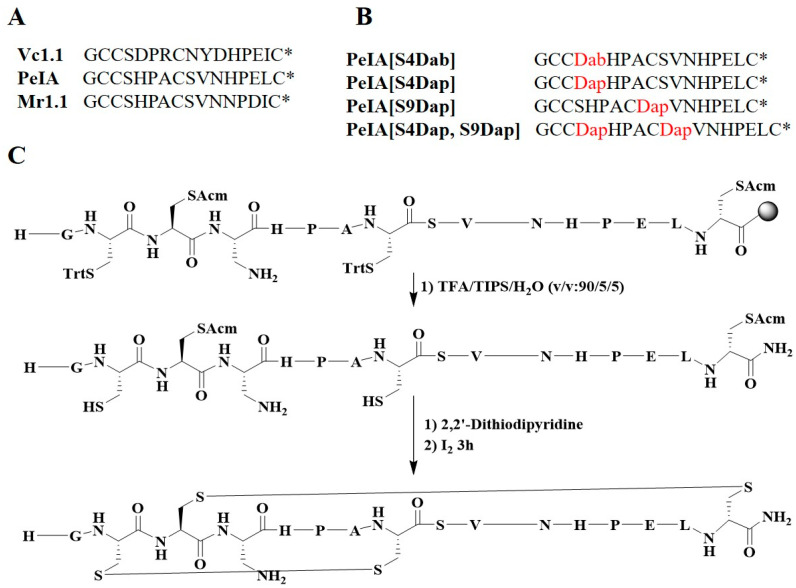
Sequences of α-conotoxins and the synthesis of the PeIA analogues. (**A**) Sequences of α-conotoxins containing two disulfide bonds (* represents the amidated C-terminus). Vc1.1, PeIA, and Mr1.1 belong to the 4/7 α-CTx subtype. (**B**) The Dab/Dap-substituted analogues of PeIA. The residue Dab/Dap is shown in red. (**C**) The synthesis of Dab/Dap-substituted analogues of PeIA using PeIA[S4Dab] as an example. The chemical structure was drawn using ChemBioDraw Ultra 14.0.

**Figure 3 marinedrugs-22-00110-f003:**
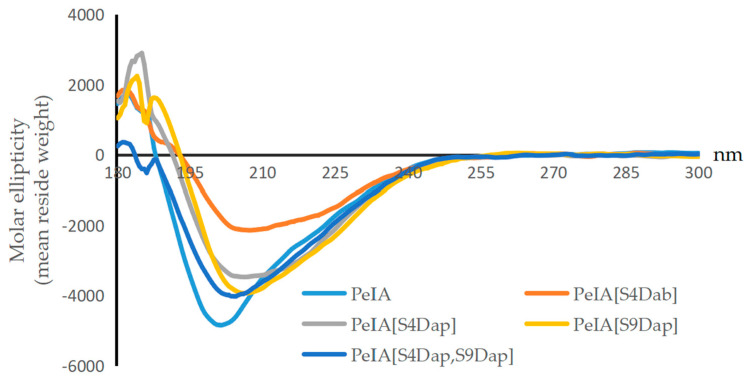
CD spectra of PeIA, PeIA[S4Dab], PeIA[S4Dap], PeIA[S9Dap], and PeIA[S4Dap, S9Dap].

**Figure 4 marinedrugs-22-00110-f004:**
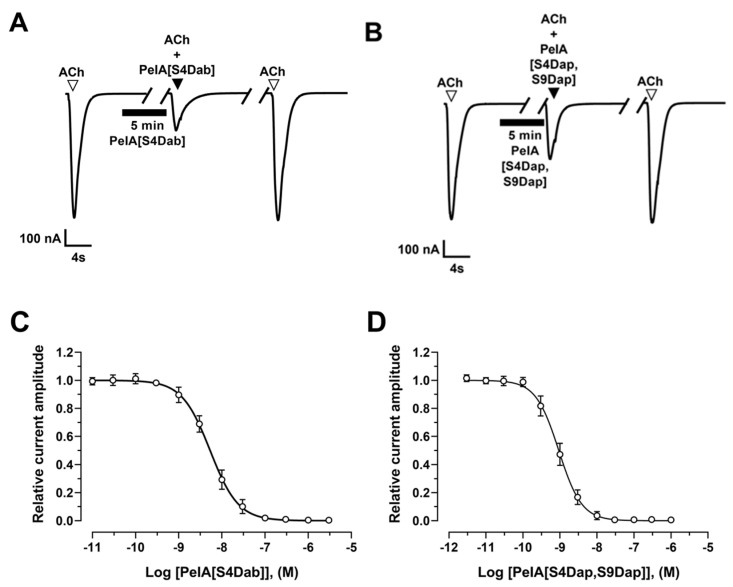
Inhibitory activities of PeIA[S4Dab] and PeIA[S4Dap, S9Dap] at hα9α10 nAChRs. Representative ACh (6 µM)-evoked currents mediated by hα9α10 nAChRs in the presence of (**A**) 10 nM PeIA[S4Dab] and (**B**) 1 nM PeIA[S4Dap, S9Dap]. ∇, ACh alone; ▼, co-application of ACh + α-CTx after a 5 min incubation (▬) with α-CTx alone; ∇, ACh alone after washout. Concentration–response relationships of the relative ACh-evoked current amplitude (mean ± SD, *n* = 6–15) mediated by hα9α10 nAChRs in the presence of (**C**) PeIA[S4Dab] and (**D**) PeIA[S4Dap, S9Dap], giving IC_50′_s of 5.3 nM (5.00–5.67; 95% confidence interval (CI)) and 0.93 nM (0.88–0.97; 95% CI), respectively. Whole-cell currents at hα9α10 were activated by 6 μM ACh.

**Figure 5 marinedrugs-22-00110-f005:**
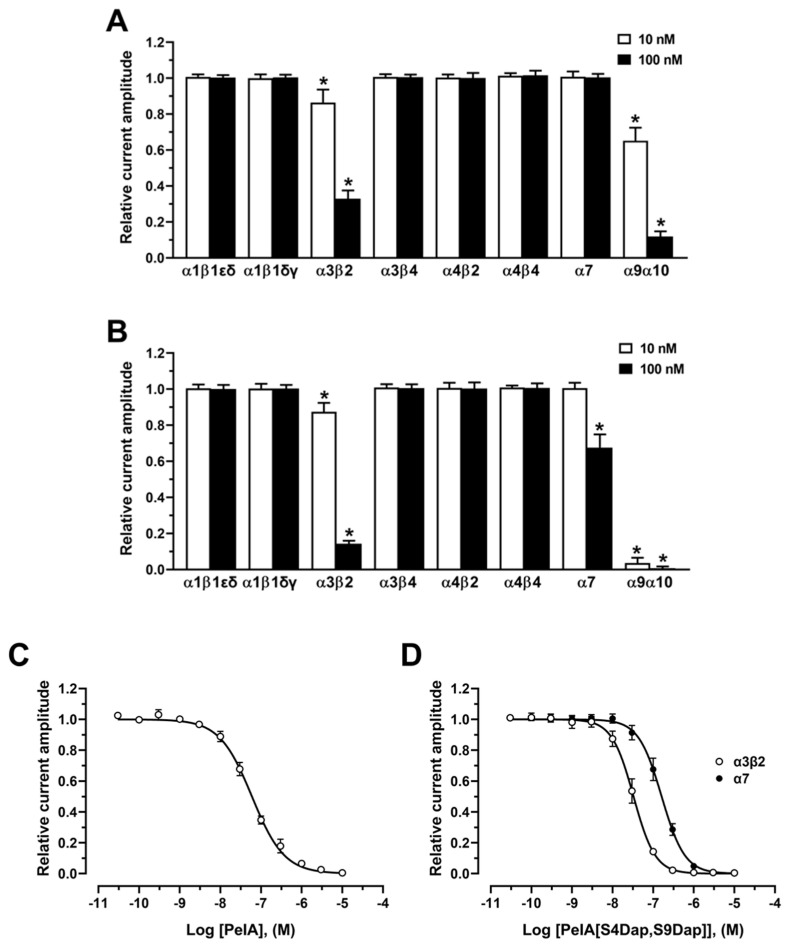
Inhibitory activities of PeIA and PeIA[S4Dap, S9Dap] at different human nAChR subtypes. Bar graphs of (**A**) PeIA and (**B**) PeIA[S4Dap, S9Dap] (10 and 100 nM) inhibition of the ACh-evoked peak current amplitude mediated by heterologous human nAChRs (mean ± SD, *n* = 5–15). * *p* < 0.05 compared to 1.0. Data for PeIA at hα9α10 were from Liang et al. [19]. (**C**) Concentration–response relationships of the relative ACh-evoked current amplitude (mean ± SD, *n* = 6–12) mediated by hα3β2 nAChRs in the presence of PeIA, giving an IC_50_ of 59.8 nM (57.0–62.8; 95% CI). (**D**) Concentration–response relationships of the relative ACh-evoked current amplitude (mean ± SD, *n* = 7–8) mediated by hα3β2 and hα7 nAChRs in the presence of PeIA[S4Dap, S9Dap], giving IC_50′_s of 32.8 nM (31.2–34.5; 95% CI) and 161.7 nM (154.0–169.8; 95% CI), respectively. Whole-cell currents at hα3β2 and hα7 were activated by 6 μM and 100 μM ACh, respectively.

**Figure 6 marinedrugs-22-00110-f006:**
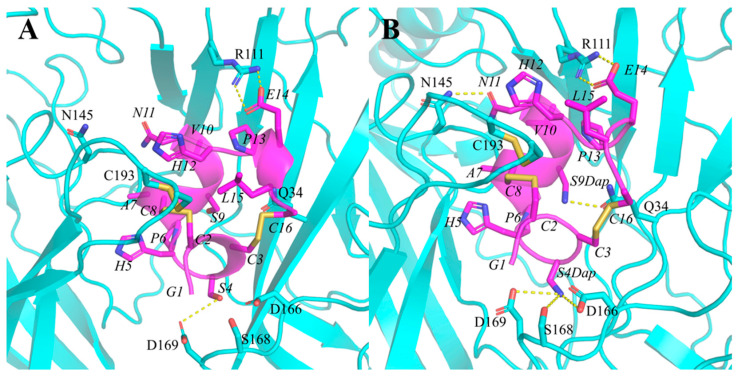
Binding modes of PeIA and PeIA[S4Dap,S9Dap] at the α9(+)α9(−) binding site. (**A**) The binding modes of PeIA at the hα9α10 nAChR. (**B**) The binding modes of PeIA[S4Dap, S9Dap] at the hα9α10 nAChR. The α9(+)α9(−) binding site was generated using the crystal structure of *A. californica* AChBP in complex with PeIA as a template (PDB Code: 5JME), as described in our previous study [9]. The parameters of non-natural amino acids were built in AMBER22. The complex models were refined using MD stimulations in AMBER 22.

**Table 1 marinedrugs-22-00110-t001:** Protocols for (A) RP-HPLC and (B) analytical RP-HPLC.

**A**	**Start**	**End**	**Time (min)**
5C_18_-MS-I (20 × 250 mm, 10 μm)	100	95	6
	95	80	14
	80	60	20
**B**	**Start**	**End**	**Time (min)**
5C_18_-MS-II (4.6 ID × 250 mm)	90	72	18
	72	50	31
	50	50	10

## Data Availability

Data are contained within the article and Appendix A.

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
