# Peer review of "Rational Design of Potent α-Conotoxin PeIA Analogues with Non-Natural Amino Acids for the Inhibition of Human α9α10 Nicotinic Acetylcholine Receptors"

_marinedrugs, 2024, doi:10.3390/md22030110_

Round 1

Reviewer 1 Report

Comments and Suggestions for Authors

In this study, the authors chemically synthesized Dab/Dap-substituted analogs of α-CTx PeIA and evaluated their activity on heterologously expressed human α9α10 nAChRs, obtaining a mutant PeIA[S4Dap, S9Dap] with a half-maximal inhibitory concentration (IC50) of 0.93 nM. The paper demonstrates certain innovation but requires further exploration. Specific issues identified are as follows:

1.     Please represent the subunit names in Figure 1 using Greek letters.

2.     The clarity of the images in the article is suboptimal; consider using clearer images.

3.     Why did the author choose to design only two mutants, specifically at positions 4 and 9 (Ser), and not explore a variety of mutations for comprehensive evaluation?

4.     In this study, the use of previous literature data for PeIA as a comparison is inappropriate. The author should supplement the study with wildtype PeIA data.

5.     Recognizing the significance of selectivity in conotoxins, preliminary research indicates PeIA's activity on various subtypes of acetylcholine receptors. While the modification of PeIA enhances its selectivity for α9α10, an assessment of its impact on other receptor activities is warranted.

6.     The data from CD testing should be presented in the results.

7.     Despite obtaining key site information through MD, why did the author not perform receptor mutations for a more in-depth investigation and validation?

8.     The font for "based" on page 134 is inappropriate.

9.     Subscripts are needed for elements like H2O in the synthetic figure; please make the necessary modifications.

Author Response

We thank the Reviewer for their helpful comments to improve the revised manuscript. The response to each question is in red font below and changes are highlighted in the revised marked manuscript.

Reviewer 1.

  1. Please represent the subunit names in Figure 1 using Greek letters.

We have made the suggested changes in Figure 1.

  1. The clarity of the images in the article is suboptimal; consider using clearer images.

We have now included clearer images.

  1. Why did the author choose to design only two mutants, specifically at positions 4 and 9 (Ser), and not explore a variety of mutations for comprehensive evaluation?

In the revised manuscript, we have included the activity of analogues PeIA[S4Dap] and PeIA[S9Dap]. In our previous studies, we used a site-specific mutagenesis strategy and designed a series of analogues to improve the binding affinity of peptides and demonstrated that non-natural amino acids Dab4 and Dap4/9 could dramatically improve the potency of the α-conotoxins Vc1.1 and Mr1.1. The [S4Dab] analogue of Vc1.1 and [S4Dap], [S9Dap], [S4Dap, S9Dap] analogues of Mr1.1 exhibited significantly greater potency. Considering the same structural subclass and the sequence similarity of PeIA, Vc1.1, and Mr1.1, and the sequence in Loop 1 and the charge distributions in Loop2 of PeIA exhibit high similarity with Mr1.1, we designed these PeIA analogues to enhance its activity. Our study provides an effective way to optimize the peptide activity and obtain a highly potent analogue of PeIA.

  1. In this study, the use of previous literature data for PeIA as a comparison is inappropriate. The author should supplement the study with wildtype PeIA data.

The literature data referenced was for PeIA at human α9α10 nAChRs. We have now included data for PeIA activity at different human nAChR subtypes (Figure 5A).

  1. Recognizing the significance of selectivity in conotoxins, preliminary research indicates PeIA's activity on various subtypes of acetylcholine receptors. While the modification of PeIA enhances its selectivity for α9α10, an assessment of its impact on other receptor activities is warranted.

Agree. We have included the activity of PeIA and PeIA[S4Dap, S9Dap] at different human nAChR subtypes (Figure 5).

  1. The data from CD testing should be presented in the results.

Agree. We have moved the CD results from the Supporting Information to the Results in the main text (Figure 3).

  1. Despite obtaining key site information through MD, why did the author not perform receptor mutations for a more in-depth investigation and validation?

Our previous study used the complex model of Mr1.1/α9α10 nAChR constructed by AMBER20 to guide the design of the analogues that may exhibit increased potency, and the results were consistent with MD. Considering the high similarity of sequences between PeIA and Mr1.1, we used the complex model Mr1.1/α9α10 nAChR as a template to construct the complex of PeIA/α9α10 nAChR and the parameters for non-natural amino acids were also made to construct the complex of the most potent analogue PeIA[S4Dap, S9Dap]/α9α10 nAChR. Our models could explain the increase in inhibitory activity. The Dap4 residue formed additional hydrogen bond interactions with the receptor, and Dap9 could also form a hydrogen bond interaction with Q34, which was different with PeIA and explained the reasons for the increased activity. Considering the MD data could interpret the experimental results with a degree of confidence, we are optimistic about the outcome. Due to time constraints, receptor mutations were beyond the scope of the present study but can be considered in future for an in-depth investigation and validation.

  1. The font for "based" on page 134 is inappropriate.

We have modified the sentence to read as “hα9α10 nAChR using MD simulations (Figure 6, S4-S6, Table S1), based on the models of Mr1.1 and Vc1.1”.

  1. Subscripts are needed for elements like H2O in the synthetic figure; please make the necessary modifications.

We have made the corrections to Figure 2.

Reviewer 2 Report

Comments and Suggestions for Authors

In this article, Li et al., further explore the use of non-natural amino acids to synthesize analogues of alpha-conotoxin PeIA with increased potency for human alpha9alpha10 nicotinic receptors. This receptor has been broadly implicated in chronic and inflammatory pain, and targeting this subtype with novel therapeutics has been proposed to be analgesic. The principal investigators are well-known experts in their field. The article is well written, and the science is sound.

Major issues

The use of non-natural amino acids to improve peptide binding and specificity is interesting, and the fact that the authors were able to synthesize a conotoxin with ~1 nM potency for human alpha9alpha10 receptors is a nice achievement. My enthusiasm for this manuscript is diminished, however, principally due to the fact that the authors have already shown that the two substitutions made in PeIA increased potency of alpha-conotoxins Vc1.1 and Mr1.1, two peptides that differ very little in sequence with respect to that of PeIA (references 9 and 19). Mr1.1, for example, only diverges significantly from PeIA by an Asn vs His at position 12, respectively. Thus, it should be no surprise that S4Dap and S9Dap increase PeIA potency for alpha9alpha10. In short, there is very little new information presented in this manuscript.

Secondly, the authors propose that the new PeIA analog has the potential to be a drug candidate for treating chronic pain. It’s difficult to agree with this statement when no selectivity profile has been provided. The argument would be supported if the authors could show (similar to Fig. 5 of ref. 9) that, at a minimum, PeIA[S4Dap,S9Dap] lacks activity on the alpha7 and alpha3beta4 subtypes to avoid significant off target effects that would occur by antagonizing these two subtypes.

Minor comments

Line 39: Only beta1-beta4 nicotinic subunits exist, please delete beta5.

Line 68: Reference 18 does not include experiments assessing PeIA on human receptors, only rat.

Figure 1A-C: The authors should state in the figure legend which PDB file was used to generate the alpha9alpha10 structure (presumably from Ref. 20) as well as the program used to generate the images. The letter a9/a10 should be the Greek symbol α.

Figure 2. Please indicate the program used to draw the chemical structures of panel C.

Figure 4: Same comment as above. Please state the PDB file and program used to generate the images. The authors are using sequence numbering that does not match the mature human alpha9 amino-acid sequence, for example D166 appears to correspond to D171 of the mature protein. Gln52 should then correspond to Gln57 of the mature protein, yet the closest Gln would be Gln53. Please provide an explanation in the figure legend for the numbering and check for accuracy. It would be very helpful if the authors could provide the MDs data in a table (similar to table 3 of Ref. 9). Also, when referenced in the text, it would be helpful, for example in the introduction, to specify which subunit (+) or (-) corresponds to which amino-acid residue (lines 83-85).

Supplementary Information

The title of the manuscript and the SI are not the same. The quality of most of the figures is poor i.e., low resolution, miniscule and unreadable font, Chinese characters, etc.

Figures S7-S9. The stated purity of near 100% appears inaccurate. Both the ESI-MS and the RP-HPLC show a small peak at ~3 min mark with a mass of 550-555 Da, indicating some type of impurity. Both peaks should be integrated and the % purity determined.

Author Response

We thank the Reviewer for their helpful comments to improve the revised manuscript. The response to each question is in red font below and changes are highlighted in the revised marked manuscript.

Reviewer 2.

Major issues

1.: The use of non-natural amino acids to improve peptide binding and specificity is interesting, and the fact that the authors were able to synthesize a conotoxin with ~1 nM potency for human alpha9alpha10 receptors is a nice achievement. My enthusiasm for this manuscript is diminished, however, principally due to the fact that the authors have already shown that the two substitutions made in PeIA increased potency of alpha-conotoxins Vc1.1 and Mr1.1, two peptides that differ very little in sequence with respect to that of PeIA (references 9 and 19). Mr1.1, for example, only diverges significantly from PeIA by an Asn vs His at position 12, respectively. Thus, it should be no surprise that S4Dap and S9Dap increase PeIA potency for alpha9alpha10. In short, there is very little new information presented in this manuscript.

In terms of peptide design, we referred to previous related studies, and considerng the same structural subclass and the sequence similarity of PeIA, Vc1.1, and Mr1.1, and the sequence in Loop 1 and the charge distributions in Loop 2 of PeIA exhibit high similarity with Mr1.1, we designed these analogues of PeIA to enhance its activity. In general, site-directed mutagenesis is used to improve the peptide potency, but a large number of analogues had no enhanced activity. Thus, we referred to peptides with similar sequences. Secondly, the obtained peptide PeIA[S4Dap, S9Dap] exhibited dramatically enhanced activity giving an IC50 of 0.93 nM. Our work provides an effective way to optimize the peptide activity and obtain a potent analogue of PeIA.

  1. Secondly, the authors propose that the new PeIA analog has the potential to be a drug candidate for treating chronic pain. It’s difficult to agree with this statement when no selectivity profile has been provided. The argument would be supported if the authors could show (similar to Fig. 5 of ref. 9) that, at a minimum, PeIA[S4Dap,S9Dap] lacks activity on the alpha7 and alpha3beta4 subtypes to avoid significant off target effects that would occur by antagonizing these two subtypes.

We have tested and included the activity of PeIA[S4Dap,S9Dap] at different human nAChR subtypes (Figure 5B). Compared to the absence of inhibition by PeIA, PeIA[S4Dap,S9Dap] had enhanced potency at human α7 nAChRs. However, in relation to alpha7-mediated analgesia, it has been proposed that activation of the α7 subtype as opposed to inhibition, is responsible for attenuation of neuropathic pain in animal models (DOI:10.1097/FBP.0b013e32834a1efb; DOI:10.1016/j.neuint.2012.09.001; DOI:10.1016/j.brainres.2019.03.016 ).

Minor comments

  1. Line 39: Only beta1-beta4 nicotinic subunits exist, please delete beta5.

We have deleted beta5.

  1. Reference 18 does not include experiments assessing PeIA on human receptors, only rat.

We have included reference 19, which describes the activity of PeIA at human nAChRs.

  1. Figure 1A-C: The authors should state in the figure legend which PDB file was used to generate the alpha9alpha10 structure (presumably from Ref. 20) as well as the program used to generate the images. The letter a9/a10 should be the Greek symbol α.

We have made the suggested changes in Figure 1 using the Greek symbol α and included the PDB file and program used.

  1. Figure 2. Please indicate the program used to draw the chemical structures of panel C.

We have included the program used in Figure 2.

  1. Figure 4: Same comment as above. Please state the PDB file and program used to generate the images. The authors are using sequence numbering that does not match the mature human alpha9 amino-acid sequence, for example, D166 appears to correspond to D171 of the mature protein. Gln52 should then correspond to Gln57 of the mature protein, yet the closest Gln would be Gln53. Please provide an explanation in the figure legend for the numbering and check for accuracy. It would be very helpful if the authors could provide the MDs data in a table (similar to table 3 of Ref. 9). Also, when referenced in the text, it would be helpful, for example in the introduction, to specify which subunit (+) or (-) corresponds to which amino-acid residue (lines 83-85).

We have included the PDB file and program used in Figure 6. The complex model of PeIA/α9α10 nAChR was built using the extracellular domain of the α9(+)α9(−) of the α9α10 nAChR constructed in our previous study (reference 9). The sequence number of α9(+) subunit starts with 0. So in our complex model, the sequence number was D166. We have included “residues D166, S168, and D169 of the α9(−) subunit” and the MD data in Table S1).

Supplementary Information

  1. The title of the manuscript and the SI are not the same. The quality of most of the figures is poor i.e., low resolution, minuscule and unreadable font, Chinese characters, etc.

The SI title has been corrected and is now the same as the manuscript. The resolution of the figures has been improved. 

  1. Figures S7-S9. The stated purity of near 100% appears inaccurate. Both the ESI-MS and the RP-HPLC show a small peak at ~3 min mark with a mass of 550-555 Da, indicating some type of impurity. Both peaks should be integrated and the % purity determined.

We have rectified the issues. We re-analyzed the purity of the sample using another analytical RP-HPLC, yet the small peak at ~3 min remains, which could be caused by the analytical RP-HPLC itself, thus the peaks are not integrated.

Reviewer 3 Report

Comments and Suggestions for Authors

Alpha-conotoxins are disulfide-rich peptides from the venom of the Cone snails and have significant potential as drug candidates and pharmacological tools. In this manuscript, the authors rational design two unnatural amino acid analogues of PeIA to improve its biological activity, and the obtained analogue PeIA[S4Dap,S9Dap] is one of the most potency conotoxins that targets α9α10 nAChRs. Considering the drug tolerance of traditional analgesics, the robust peptide associated with α9α10 nAChRs is urgently needed, and this work affords a practicable method to optimize the activity of the peptide and thus may address the ongoing opioid crisis. It is important for the medicinal chemists working on drug development from this work. So, I agree to its publication after minor revisions.

1. Abbreviations should be used except for the first time. The “nicotinic acetylcholine receptors” and “IC50 in the introduction section should be abbreviated.

2. Please describe the composition of the nicotinic acetylcholine receptor in the introduction section. (α1–α10, β1–β5, γ, δ and ε).

3. The “IC50” in 2.2, Figure 3, 4.7(line 270) and the Discussion parts should be subscript. Please check the table 1, in which the ‘5C18-MS-I’ should be modified ‘18’ should be subscript.

4. Please check the place of ‘Conclusion’ part. Generally, it is after the discussion rather than after the method part.

5. The references maybe more fully cited on relevant to this research.

6. The methodology was not fully understood because the supporting information could not be downloaded.

Author Response

We thank the Reviewer for their helpful comments to improve the revised manuscript. The response to each question is in red font below and changes are highlighted in the revised marked manuscript.

Reviewer 3.

  1. Abbreviations should be used except for the first time. The “nicotinic acetylcholine receptors” and “IC50” in the introduction section should be abbreviated.

We have made the suggested changes.

  1. Please describe the composition of the nicotinic acetylcholine receptor in the introduction section. (α1–α10, β1–β5, γ, δ and ε).

We have described the composition of the nicotinic acetylcholine receptors in the 2nd paragraph of the Introduction.

  1. The “IC50” in 2.2, Figure 3, 4.7(line 270) and the Discussion parts should be subscript. Please check the table 1, in which the ‘5C18-MS-I’ should be modified ‘18’ should be subscript.

We have subscripted the numbers for IC50 and C18 in 5C18-MS-I (Table 1).

4.: Please check the place of ‘Conclusion’ part. Generally, it is after the discussion rather than after the method part.

As per the journal’s format, the Conclusions comes after the Materials and Methods section.

  1. The references maybe more fully cited on relevant to this research.

We have included relevant references.

  1. The methodology was not fully understood because the supporting information could not be downloaded.

Supporting information has been included.

Round 2

Reviewer 1 Report

Comments and Suggestions for Authors

The authors have carefully revised the manuscript in accordance with the reviewer's advice and agree with its publication.

Reviewer 2 Report

Comments and Suggestions for Authors

The additional testing on other nAChR subtypes is appreciated.